# Solution-Processed Smooth Copper Thiocyanate Layer with Improved Hole Injection Ability for the Fabrication of Quantum Dot Light-Emitting Diodes

**DOI:** 10.3390/nano12010154

**Published:** 2022-01-01

**Authors:** Ming-Ru Wen, Sheng-Hsiung Yang, Wei-Sheng Chen

**Affiliations:** 1Institute of Lighting and Energy Photonics, College of Photonics, National Yang Ming Chiao Tung University, Tainan 71150, Taiwan; lg0895623@gmail.com; 2Opulence Optronics Co., Ltd., Hsinchu 30091, Taiwan; nabilies.00g@g2.nctu.edu.tw

**Keywords:** molecular doping, copper thiocyanate, hole injection layer, quantum dot, light-emitting diode

## Abstract

Copper thiocyanate (CuSCN) has been gradually utilized as the hole injection layer (HIL) within optoelectronic devices, owing to its high transparency in the visible range, moderate hole mobility, and desirable environmental stability. In this research, we demonstrate quantum dot light-emitting diodes (QLEDs) with high brightness and current efficiency by doping 2,3,5,6-tetrafluoro-7,7,8,8-tetracyanoquinodimethane (F4TCNQ) in CuSCN as the HIL. The experimental results indicated a smoother surface of CuSCN upon F4TCNQ doping. The augmentation in hole mobility of CuSCN and carrier injection to reach balanced charge transport in QLEDs were confirmed. A maximum brightness of 169,230 cd m^−2^ and a current efficiency of 35.1 cd A^−1^ from the optimized device were received by adding 0.02 wt% of F4TCNQ in CuSCN, revealing promising use in light-emitting applications.

## 1. Introduction

Since the first electroluminescent device based on colloidal quantum dots (QDs) was reported in 1994 [1], many research teams have demonstrated the use of QDs as the emissive layer to fabricate quantum dot light-emitting diodes (QLEDs) [2,3,4,5,6,7,8,9]. QLEDs are regarded as one promising choice for next generation displays and solid-state lighting because of their unique properties, such as high brightness, color tunability, low cost, and the ability to emit very pure light. Cadmium selenide (CdSe) is one of the most common QD materials, which possesses superior emissive efficiency and narrow size distribution. Over 100 cd A^−1^ efficient devices with inverted tandem structure have been demonstrated [10], promoting mass production and commercialization of QLEDs in lighting and display areas.

The multi-layered QLEDs are usually constructed with the configuration of anode/hole injection layer (HIL)/hole transport layer (HTL)/emission layer (EML)/electron transport layer (ETL)/electron injection layer (EIL)/cathode, so-called normal architecture which is similar to that of organic light-emitting diodes (OLEDs). Moreover, both QLEDs and OLEDs can be fabricated via the solution process with low cost. In the QLED structure, inorganic zinc oxide (ZnO) nanoparticles (NPs) have been widely used as the ETL so far because it has high transmittance in the visible range, high electron mobility, and a matched energy level with the EML and/or cathode [11,12,13]. The HIL/HTL materials should ideally possess the following properties: (1) high transparency across the full visible spectrum [14,15], (2) suitable work function to allow hole injection/transport and blocking of electrons to prevent leakage current [16], and (3) feasible deposition technique to achieve large-area manufacture and low production cost [14,17,18]. Several classic polymers have been reported as the HIL/HTL in QLED fabrication, including poly(3,4-ethylenedioxythiophene):poly(4-styrenesulfonate) (PETDOT:PSS), poly[*N*-(4-butylphenyl)-*N*’,*N*’’-diphenylamine] (poly-TPD), poly(9,9-dioctylfluorene- co-*N*-(4-butylphenyl)diphenylamine)] (TFB), and poly(*N*-vinylcarbazole) (PVK) [19,20,21]. Despite the rapid development in QLEDs, imbalanced electron and hole transport often occurs that restricts device performance. In contrast to ZnO, the abovementioned hole transport polymers have much lower conductivity; besides, the energy barrier between the HTL and QDs are usually larger than that at the QDs/ETL interface, causing imbalanced carriers transport and restricted device performance [22]. Thus, many research groups have been devoted to improving charge balance through prohibition of electron injection or augmentation of hole transport ability. X. Dai et al. introduced an ultrathin insulating poly(methyl methacrylate) (PMMA) layer between the QDs and ETL to optimize charge balance in devices [23], resulting in a record-high external quantum efficiency (EQE) of 20.5% and a long operation lifetime of more than 100,000 h at 100 cd m^−2^. S. Wang et al. reported that doping Mg^2+^ in ZnO to form solution-processed zinc magnesium oxide (ZnMgO) films can slow down the electron mobility to reach a balanced injection of electrons and holes toward a better device performance [11]. The peak current efficiency was enhanced from 3.74 to 13.73 cd A^−1^, revealing a 3.67-fold enhancement compared to the reference device using the undoped ZnO as the ETL. L. Wang et al. also inserted the ZnMgO NPs between ZnO ETL and InP emissive layer to reduce exciton quenching and electric current [24]. A 2.36-fold enhancement of current efficiency than the reference device without the ZnMgO interlayer was achieved. The second approach is to augment hole transport ability of the HTL to bring electron-hole charge balance in QLEDs. Our group has blended a commercial dispersing additive BYK-P105 into PEDOT:PSS to improve hole mobility and to reach balanced charge transport in our QLEDs [25]. The best QLED exhibited a low turn-on voltage of 3.8 V, a current efficiency of 27.2 cd A^−1^, and a very high brightness of 139,909 cd m^−2^. Apart from PEDOT:PSS, the PVK layer has also been modified to improve hole transport ability as well as QLED performance. Y. Shi et al. doped lithium bis (trifluoromethylsulfonylamine) (Li-TFSI) into PVK to improve the hole mobility and then reach balanced carrier injection, which can contribute to higher performance of the QLEDs [26]. With an optimal doping ratio of 3.0 wt% Li-TFSI in PVK, the QLED received a current efficiency of 15.5 cd A^−1^ and a very high EQE value of 11.46%, which was almost two-fold higher than the one using pristine PVK as the HTL (EQE = 6.58%).

Until now, PEDOT:PSS is the most popular hole injection material for fabricating QLEDs with high luminance and efficiency [21,22,23,24,25,26]. However, it is reported that the conductivity of PEDOT:PSS film is highly connected to the acid treatments which can deteriorate indium tin oxide (ITO) anodes and therefore decrease device performance [18,27]. Recently, many researchers have been dedicated to utilizing inorganic metal oxides as the HIL, such as nickel oxide (NiO), molybdenum oxide (MoO_3_), tungsten oxide (WO_3_), copper oxide (CuO), and vanadium oxide (V_2_O_5_) [28,29,30,31,32], due to their high carrier mobility and excellent thermal stability in contrast to organic materials. However, there still exist some drawbacks in the preparation of inorganic metal oxide layers for QLEDs. In general, metal oxide films via the sol-gel process require high-temperature calcination to achieve suitable transparency and conductivity [20], which may increase the resistance of ITO electrode, and thus lower the performance of QLEDs [33]. On the other hand, metal oxide NPs dispersions are synthesized to deposit thin films with low-temperature annealing, which can still face difficulties in aggregates and film-forming properties of NPs. To avoid those drawbacks, we turn our attention to other materials exclusive of metal oxide HILs. After surveying the previous literature, we realize that copper thiocyanate (CuSCN) possesses several advantages and can act as a good HIL in optoelectronic devices, including high stability in an ambient atmosphere, high transparency in the whole visible range, moderate hole mobility of 0.01–0.1 cm^2^ V^−1^ s^−1^, and deep valence band level (*E*_VB_) around 5.3 eV [17,34]. CuSCN has been utilized in perovskite solar cells, thin film transistors, QLEDs, and OLEDs [17,20,33,34,35,36], and can be deposited into thin films by the solution process with a low-temperature drying. In this research, we adopt CuSCN as the HIL for the fabrication of QLEDs, and a strong electron-withdrawing molecule 2,3,5,6-tetrafluoro-7,7,8,8-tetracyanoquinodimethane (F4TCNQ) was introduced to further improve hole mobility of CuSCN by *p*-type doping. F4TCNQ has been incorporated into PEDOT:PSS or a star-shaped hole transport material to increase conductivity of the above HTL and to reduce the hole injection barrier [37,38]. Herein, we demonstrate high-efficiency QLEDs by using F4TCNQ-doped CuSCN as the HIL for the first time. Green QLEDs with the configuration of ITO/F4TCNQ-doped CuSCN/PVK/CdSe QDs/polyethylenimine ethoxylated (PEIE)/ZnO NPs/PEIE/LiF/Al were fabricated and evaluated. PEIE was selected as the interfacial layer to modify contact between ZnO NPs and CdSe QDs in our devices. The best QLED achieved a low turn-on voltage of 3.6 V, a current efficiency of 35.1 cd A^−1^, and a high brightness of 169,230 cd m^−2^. Our results reveal a new opportunity toward QLED fabrication with high performance.

## 2. Materials and Methods

### 2.1. Materials

ITO glass substrates (7 Ω/square) were purchased from Aimcore Technology Co., Ltd., Hsinchu, Taiwan. CuSCN (purity 99%), diethyl sulfide (purity 98%), and 37 wt% PEIE solution in water were bought from Sigma-Aldrich (St. Louis, MO, USA). F4TCNQ and PVK were purchased from TCI (Tokyo, Japan). CdSe QDs were provided by Opulence Optronics Co., Ltd., from Hsinchu, Taiwan. Other reagents and solvents were bought from Alfa Aesar (Ward Hill, MA, USA), ACROS, (Geel, Belgium) and TEDIA (Fairfield, OH, USA) and used without further purification. The preparation of ZnO NPs was referred to in the previous literature [25].

### 2.2. Device Fabrication

Regular QLEDs with the structure of ITO/CuSCN/PVK/CdSe QDs/PEIE/ZnO NPs/PEIE/LiF/Al were fabricated. The ITO substrates were cleaned sequentially with detergent, DI water, acetone, and IPA under ultrasonication for 30 min each, followed by nitrogen purge and ultraviolet-ozone exposure for 20 min. The CuSCN was dissolved in diethyl sulfide at a concentration of 30 mg mL^−1^ at room temperature with overnight stirring. For *p*-doping, 0.01, 0.02, and 0.03 wt% of F4TCNQ relative to CuSCN were introduction to CuSCN solution. The CuSCN layer without and with F4TCNQ was spin-cast onto the ITO substrate at 5000 rpm for 30 s and baked at 60 °C for 20 min in air. The substrates were then transferred into a nitrogen-filled glove box. PVK (in chlorobenzene, 8 mg mL^−1^) was deposited by spin coating on top of the CuSCN layer at 3000 rpm for 30 s, followed by heating at 150 °C for 20 min. CdSe QDs (in *n*-octane, 12 mg mL^−1^) were spin coated into thin film on the PVK layer at 2000 rpm for 30 s and heated at 150 °C for 30 min. A thin layer of PEIE was spin-cast from its 0.4 wt% solution in 2-ethoxyethanol on CdSe QDs at 6000 rpm for 30 s and dried at 110 °C for 20 min. ZnO NPs were spin-cast on top of the PEIE layer at 2000 rpm for 30 s, followed by heating at 150 °C for 30 min. A second PEIE layer was deposited on ZnO NPs for interfacial modification. Finally, 0.5 nm of LiF and 100 nm of aluminum electrodes were thermally evaporated at a base pressure of ~10^−6^ Torr. The active area of each device was 1 mm^2^ for device evaluation or 4 mm^2^ for snapshot of driving devices.

### 2.3. Characterization Methods

The cross-sectional micrographs of QLEDs were investigated with an ultrahigh resolution ZEISS Crossbeam scanning electron microscope (SEM) (Oberkochen, Germany). The morphology and size of CdSe QDs were examined with a JEOL JEM-1400 transmission electron microscope (TEM) (Tokyo, Japan). The surface morphology and roughness of CuSCN layers were verified using a Bruker Innova atomic force microscope (AFM) (Billerica, MA, USA). The absorption and photoluminescence (PL) spectra of CdSe QDs were recorded with a Princeton Instruments Acton 2150 spectrophotometer (Acton, MA, USA) equipped with a Xe lamp as the light source. The X-ray photoelectron spectroscopy (XPS) measurements were conducted by a Thermo KAlpha XPS instrument (Waltham, MA, USA) for elemental composition analysis of CuSCN without and with F4TCNQ. All peaks in XPS spectra were calibrated using the C *1 s* peak (adventitious carbons) at 284.4 eV. The ultraviolet photoelectron spectroscopy (UPS) measurements for the pristine and F4TCNQ-doped CuSCN were performed on a PHI 5000 VersaProbe Ⅲ (Kanagawa, Japan). A He I (hν = 21.22 eV) discharge lamp was used as the excitation source. The current density-voltage characteristics of hole- and electron-only devices were measured using an Agilent 4155C semiconductor parameter analyzer (Santa Clara, CA, USA). The performance and electroluminescent spectra of QLEDs were recorded using an Agilent 4155C semiconductor parameter analyzer and an Ocean Optics USB2000+ spectrometer (Orlando, FL, USA).

## 3. Results and Discussion

### 3.1. Characterization of CuSCN Layers and CdSe QDs

Figure 1 shows the AFM topographic images of the pristine and F4TCNQ-doped CuSCN films, revealing uniform and compact surfaces. The average roughness (Ra) of the pristine CuSCN film was estimated to be 2.50 nm, and it was decreased to 1.92, 1.56, and 1.21 nm when 0.01, 0.02, and 0.03 wt% F4TCNQ was incorporated in CuSCN, respectively. Apparently, the addition of F4TCNQ is helpful to smooth the surface of CuSCN. It is believed that the smooth and compact surface is favorable for reducing leakage current [39], which is expected to improve the device performance. The top-view SEM micrographs of the pristine and F4TCNQ-doped CuSCN films are displayed in Appendix A, revealing less pinholes on the CuSCN surface upon F4TCNQ doping. The SEM and AFM results showed that the F4TCNQ-doped CuSCN film has a smoother and more compact surface than the pristine one.

To investigate the energy levels of the pristine and F4TCNQ-doped CuSCN layer, the corresponding UPS spectra were measured and demonstrated in Appendix A. The work function, which represents the energy difference between *the* vacuum energy level and *Fermi level (E_F_),* was obtained by subtracting the high-binding energy cutoff (*E*_cutoff_) from the incident He I photon energy (hν = 21.22 eV). Appendix A shows that *E*_cutoff_ significantly decreased from 16.34 eV for the pristine CuSCN thin film to 16.1 eV for the F4TCNQ-doped CuSCN thin film, showing a ∼0.24 eV variation in *E*_cutoff_. Therefore, the *E*_F_ of the CuSCN thin films without and with F4TCNQ-doping are calculated to be −4.88 and −5.12 eV, respectively. It is known that the fluoro and cyano functional groups in F4TCNQ are the electron-withdrawing groups in nature, which results in p-doping for CuSCN and a downward shift in energy level [40]. A fluorinated fullerene derivative C_60_F_48_ has also been utilized as the molecular dopant to adjust the energy levels of CuSCN [35]. The addition of C_60_F_48_ shifts the E_F_ of CuSCN toward the *E*_VB_, i.e., *p*-doping process. The *E*_VB_ can be deduced from the equation (Equation (1)),
*E*_VB_ = *E*_F_ + *E*_onset_(1)
where *E*_onset_ represents the onset in the valence-band edge [19]. The *E*_VB_ of the pristine and F4TCNQ-doped CuSCN thin film are −5.3 and −5.42 eV, respectively. The *E*_VB_ of our pristine CuSCN film is similar to the previous report [40], while the F4TCNQ-doped CuSCN shows a slightly lower *E*_VB_ level due to different doping ratio. In our device, a PVK layer was inserted between CuSCN and CdSe QDs to serve as the HTL. The highest-occupied molecular orbital (HOMO) of PVK is reported to be −5.8 eV [23,41]. In consequence, a smaller energy barrier exists at the interface between the F4TCNQ-doped CuSCN and PVK as compared with the pristine CuSCN, which promotes hole injection from ITO to PVK HTL through F4TCNQ-doped CuSCN HIL. In other words, F4TCNQ doping slightly alters the energy levels of CuSCN to reduce the energy barrier at the CuSCN/PVK interface.

To verify electron transfer from CuSCN to F4TCNQ, core-level peaks for the pristine and F4TCNQ-doped CuSCN films were analyzed by XPS technique and corresponding spectra are displayed in Figure 2. For the pristine CuSCN, the XPS spectrum in the Cu *2p* region shows a distinct doublet of Cu *2p*_3/2_ and Cu *2p*_1/2_ at 932.5 and 952.4 eV in Figure 2a, respectively, which corresponds to the Cu(I) chemical state [42]. A large signal at 163.0 eV dominates the S *2p* spectrum in Figure 2b, which is assigned to a sulfur atom in a ^–^S–C form [43]. Figure 2c shows the C *1s* core-level spectrum that reveals a main band centered at 285.6 eV (SCN bond) and a shoulder band at 284.4 eV (adventitious carbons) [35]. A single N *1s* peak is found at 398.2 eV in Figure 2d, which is attributed to the C≡N bonding [44]. Moreover, the core-level binding energy of the F4TCNQ-doped CuSCN shows an obvious shift of 0.3–0.5 eV compared with the pristine one. The shift toward high binding energy conforms to the phenomena of *p*-type doping and effective electron transfer from CuSCN to F4TCNQ [35,40]. Besides, an additional signal for F *1s* was detected at 687.2 eV for the F4TCNQ-doped CuSCN [40], revealing successful F4TCNQ doping in CuSCN.

The TEM image of CdSe QDs is displayed in Appendix A and an average diameter of 10 nm is observed. The PL emission spectra and UV–vis absorption of CdSe QDs in the solution state are displayed in Appendix A. It is seen that our CdSe QDs showed a pure green emission at 525 nm with a narrow full width at half-maximum (FWHM) value of 28 nm.

### 3.2. Device Evaluation

The illustration of the QLED based on the F4TCNQ-doped CuSCN is shown in Figure 3a, which is fabricated with the structure of ITO/F4TCNQ-doped CuSCN/PVK/CdSe QDs/PEIE/ZnO NPs/PEIE/LiF/Al. The energy level diagram of the device is displayed in Figure 3b. We chose CuSCN as the HIL since its VB is close to the work function of ITO. The E_VB_ of CuSCN was deduced from the abovementioned UPS analysis, and corresponding conduction band level (*E*_CB_) was obtained from the following equation (Equation (2)),
*E*_CB_ = *E*_VB_ + *E*_g_(2)
where *E*_g_ represents the energy bandgap from the absorption spectra to be 3.88 eV. Therefore, the *E*_CB_ was calculated to be −1.42 and −1.54 eV, respectively, for the pristine and F4TCNQ-doped CuSCN films. PVK was used as the HTL between the CuSCN HIL and CdSe QDs emissive layer for hole transportation. The HOMO and LUMO of PVK are reported to be −5.8 and −2.2 eV, respectively [23,41]. In addition, PVK possesses good electron-blocking capability due to the high-lying LUMO level. As for the active layer, the *E*_VB_ and *E*_CB_ of the CdSe QDs are referred to our previous literature to be −6.6 and −4.4 eV [25], respectively. About the ETL, the *E*_CB_ of the PEIE-modified ZnO is −2.9 eV [45], while LiF/Al was evaporated as the cathode with a work function of −2.8 eV [46], indicative of a small energy barrier for electron injection from LiF/Al electrode to ZnO/PEIE. The cross-sectional SEM image of the whole device is shown in Figure 3c. The thicknesses of the F4TCNQ-doped CuSCN/PVK, CdSe QDs/PEIE, ZnO/PEIE, and LiF/Al are estimated to be 50, 35, 50, and 100 nm, respectively.

To evaluate the utility of F4TCNQ-doped CuSCN in our devices, the brightness–voltage, current density–voltage, current efficiency–current density, and EQE–voltage characteristics of all QLEDs are depicted in Figure 4a–d, respectively. The control device using the undoped CuSCN as the HIL exhibited a maximum brightness (*L*_max_), current efficiency (CE_max_), and EQE of 125,129 cd m^−2^, 20.25 cd A^−1^, and 4.57%, respectively. The best device performance was achieved by adding 0.02 wt% of F4TCNQ in CuSCN and the corresponding *L*_max_, CE_max_, and EQE were significantly augmented to 169,230 cd m^−2^, 35.1 cd A^−1^, and 7.91%. Compared with the undoped CuSCN, the brightness, current efficiency, and EQE of the 0.02 wt% F4TCNQ-doped device are improved by 35.2%, 73.3%, and 73.1%, respectively. Besides, the *L*_max_, CE_max_, and EQE of the QLEDs based on 0.01 and 0.03 wt% of F4TCNQ were substantially reduced to 142,752–154,266 cd m^−2^, 25.49–28.29 cd A^−1^, and 5.29–6.35%, respectively. Thus, the optimized doping ratio of F4TCNQ in CuSCN was found to be 0.02 wt%. In addition, the device based on 0.02 wt% F4TCNQ dopant exhibited the lowest turn-on voltage of 3.6 V among all QLEDs, which is 0.43 V lower than the one with the undoped CuSCN HIL, suggesting that F4TCNQ doping can effectively improve carrier injection. Figure 4e reveals the snapshot of the QLED driven at 10 V using 0.02 wt% F4TCNQ-doped CuSCN thin film as the HIL and its electroluminescence (EL) spectrum located at 527 nm, presenting a bright green emission. The device performance from all QLEDs based on the pristine and F4TCNQ-doped CuSCN with different concentration of 0.01, 0.02, and 0.03 wt% is summarized in Table 1. It is seen that the performance of CuSCN-based QLEDs was enhanced with increasing F4TCNQ concentration, and the best performance was achieved with 0.02 wt% F4TCNQ. The device performance was then decreased with a higher F4TCNQ doping ratio of 0.03 wt%. We speculate that imbalanced electron-hole transport might occur when overdosed F4TCNQ was incorporated. Besides, high doping concentration has been reported to cause structural changes and energetic disorder in the host semiconductor, which reduces the efficiency and brightness of the QLEDs [22,35]. The performance comparison of green QLEDs from the previous literature works and this research is listed in Table 2 [5,6,7,9,19,20,25,30,46,47]. Our optimized device, using F4TCNQ-doped CuSCN as the HIL, was measured to exhibit higher brightness and efficiency at the same time among these studies. Therefore, doping F4TCNQ with CuSCN is proven to be a useful method to improve the performance of QLEDs.

To verify the improved hole transport ability of F4TCNQ-doped CuSCN and to examine whether charge balance is achieved, two hole-only devices with the configuration of ITO/pristine or F4TCNQ-doped CuSCN/Al and one electron-only device of ITO/PEIE/ZnO NPs/PEIE/LiF/Al were fabricated and compared. The current–voltage curves of the above three devices are plotted in Figure 5, revealing that the device based on the F4TCNQ-doped CuSCN film possessed a higher current and better transport of holes than that of the pristine CuSCN device. Moreover, the current of the F4TCNQ-doped CuSCN film device was closer to that of the electron-only device, indicative of more balanced electron-hole transport. The hole mobility (μ_h_) of the CuSCN thin films without and with F4TCNQ was deduced by the space-charge-limited current model with the equation (Equation (3)) [48].
J = (9/8)Ԑ_r_Ԑ_0_μ_h_V^2^/L^3^(3)

The plot ln(JL^3^/V^2^) versus electric field (V/L)^0.5^ is depicted in Figure 5b and the μ_h_ of the pristine CuSCN film is calculated to be 3.67 × 10^−2^ cm^2^ V^−1^ s^−1^, which is in the range of the reported values (0.01–0.1 cm^2^ V^−1^ s^−1^) in the previous literature [14]. Moreover, the μ_h_ of the F4TCNQ-doped CuSCN film was slightly increased to 5.53 × 10^−2^ cm^2^ V^−1^ s^−1^, suggesting positive effect of F4TCNQ doping on hole mobility and carrier transport of the CuSCN film. It is concluded that more balanced carrier transport was achieved by doping F4TCNQ into CuSCN as the HIL to obtain enhanced device performance, as shown in Figure 4 and Table 1. The smaller current densities of QLEDs in Figure 4b indicate that electron transport is suppressed. Similar observation has also been reported in the previous literature [22].

## 4. Conclusions

In this research, the F4TCNQ-doped CuSCN film was successfully utilized as the HIL in QLEDs via the solution process. The SEM and AFM results showed that the F4TCNQ-doped CuSCN was a smoother and more compact surface than the pristine one. The UPS results proved that F4TCNQ doping slightly alters the energy levels of CuSCN to reduce the energy barrier at the CuSCN/PVK interface. The XPS results confirmed *p*-type doping of CuSCN by F4TCNQ. Our results indicate that F4TCNQ doping can bring positive effects on properties of CuSCN, including smoother surface, higher hole mobility, and more balanced carrier transport, as compared with the pristine one. The best device exhibited a low turn-on voltage of 3.6 V, a current efficiency of 35.1 cd A^−1^, and a maximum brightness of 169,230 cd m^−2^. Compared to currently used HILs such as organic PEDOT:PSS or inorganic nickel oxide, F4TCNQ-doped CuSCN has better stability against moisture and can be processed at lower temperatures, respectively. Our results suggest that F4TCNQ-doped CuSCN is a potential HIL candidate for QLED fabrication.

## Figures and Tables

**Figure 1 nanomaterials-12-00154-f001:**
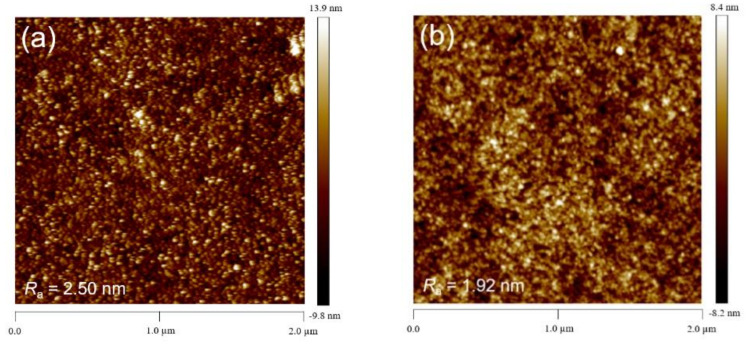
AFM topographic images of the (**a**) pristine and (**b**) 0.01, (**c**) 0.02, and (**d**) 0.03 wt% F4TCNQ-doped CuSCN films.

**Figure 2 nanomaterials-12-00154-f002:**
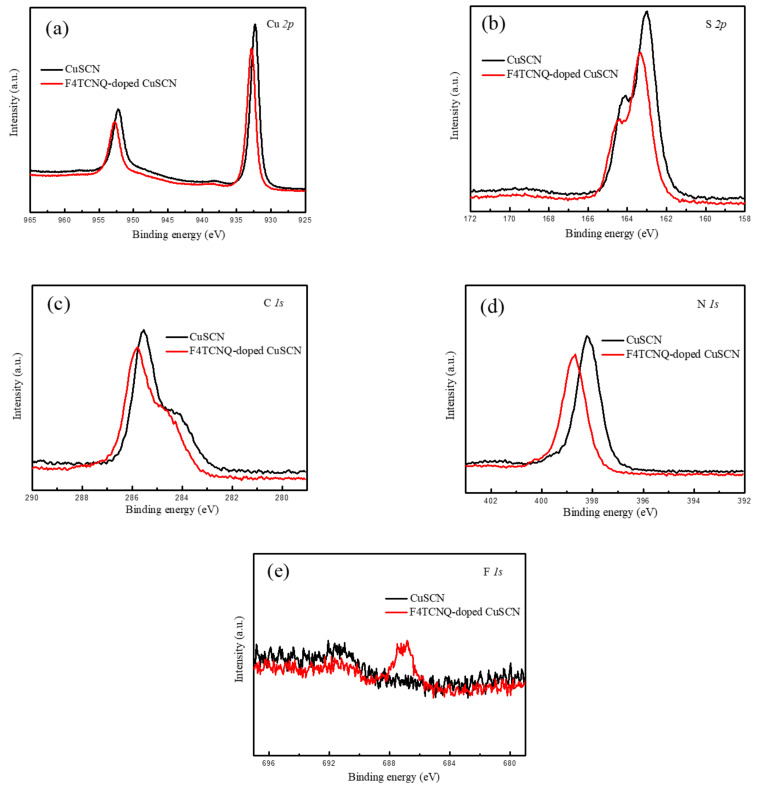
XPS spectra of the pristine and F4TCNQ-doped CuSCN films for the core level of (**a**) Cu *2p*, (**b**) S *2p*, (**c**) C *1s*, (**d**) N *1s*, and (**e**) F *1s*.

**Figure 3 nanomaterials-12-00154-f003:**
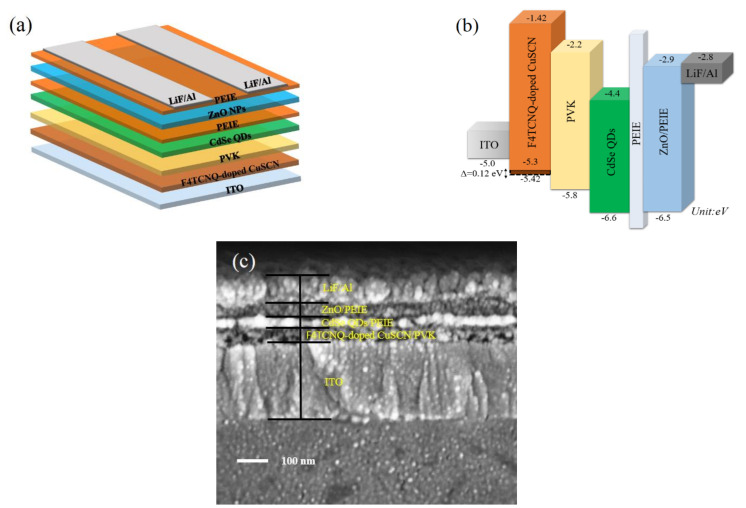
(**a**) Device architecture, (**b**) energy-level diagram, and (**c**) cross-sectional SEM image of the QLED.

**Figure 4 nanomaterials-12-00154-f004:**
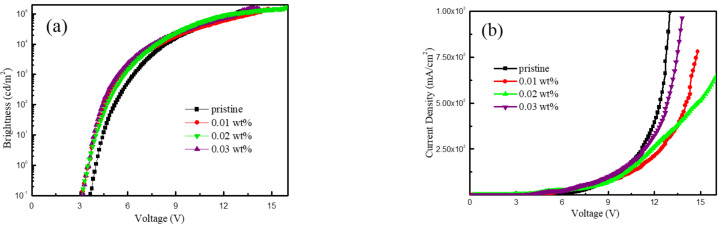
(**a**) Brightness–voltage, (**b**) current density–voltage, (**c**) current efficiency–current density, (**d**) EQE–voltage, and (**e**) EL spectrum and snapshot of the driving device based on 0.02 wt% F4TCNQ-doped CuSCN at 10 V.

**Figure 5 nanomaterials-12-00154-f005:**
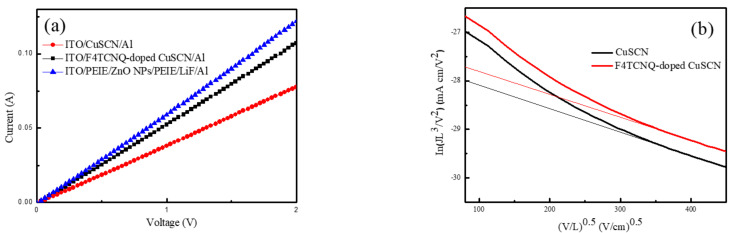
(**a**) Current-voltage curves of the hole-only and electron-only devices. (**b**) Hole mobility of the pristine and F4TCNQ-doped CuSCN versus electric field (V/L)^0.5^.

**Table 1 nanomaterials-12-00154-t001:** Device performance of all QLEDs based on the pristine and F4TCNQ-doped CuSCN as the HIL.

Doping Ratio (wt%)	V_on_ ^a^ (V)	*L*_max_ (cd m^−2^ @V)	CE_max_ (cd A^−1^ @V)	EQE_max_ (% @V)
0	4.03	125,129 @13.2	20.25 @10.0	4.57 @10.0
0.01	3.66	142,752 @14.8	28.29 @10.0	6.35 @10.0
0.02	3.60	169,230 @16.0	35.10 @9.90	7.91 @9.90
0.03	3.60	154,266 @13.8	25.49 @10.6	5.29 @10.6

^a^ defined as the operating voltage when the brightness reached 1 cd m^−2^.

**Table 2 nanomaterials-12-00154-t002:** Comparison of green QLED performance based on different device structures.

Device Structure	*V*_th_ (V)	*L*_max_ (cd m^−2^)	CE_max_ (cd A^−1^)	Ref.
ITO/F4TCNQ-doped CuSCN/PVK/CdSe QDs/PEIE/ZnO NPs/PEIE/LiF/Al	3.6	169,230	35.1	This work
ITO/ZnO NPs/core–shell CdSe–ZnS QDs/PEIE/Poly-TPD/Cu:TPA/Al	2.5	69,440	78.9	[5]
ITO/ZnO NPs/CdSe@ZnS QDs/CBP/MoO3/Al	2.4	218,800	19.2	[6]
ITO/PEDOT:PSS/TFB/core–shell CdSe–ZnSe QDs/ZnO NPs/Al	N/A	614,000	N/A	[7]
ITO/ZnMgO/QDs/PVK/GO-doped PEDOT:PSS/Al	4.1	142,165	30.4	[9]
ITO/ZnO NPs/CdSe@ZnS-ZnS QDs/PEIE/Poly-TPD/MoOx/Al	3.1	110,205	65.3	[19]
ITO/CuSCN/PVK/CdSe QDs/TPBi/LiF/Al	3.4	146,700	28.4	[20]
ITO/PEDOT:PSS + BYK-P105/PVK/CdSe QDs/ZnO NPs/PEIE/LiF/Al	3.8	139,909	27.2	[25]
ITO/WO_3_ NPs/Poly-TPD/CdSe QDs/ZnO NPs/Al	3.0	21,300	4.4	[30]
ITO/PEDOT:PSS/Poly-TPD/CdZnSeS QDs/TPBi/LiF/Al	3.5	<10,000	5.2	[46]
ITO/ZnO NPs/PEIE/core–shell CdSe–ZnS QDs/PVK/PEDOT:PSS/Al	4.0	35,860	59.1	[47]

## Data Availability

Not applicable.

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
