# Peer review of "Solution-Processed Smooth Copper Thiocyanate Layer with Improved Hole Injection Ability for the Fabrication of Quantum Dot Light-Emitting Diodes"

_nanomaterials, 2022, doi:10.3390/nano12010154_

Round 1

Reviewer 1 Report

The manuscript presents the advantages of adding 2,3,5,6-tetrafluoro- 13 7,7,8,8-tetracyanoquinodimethane (F4TCNQ) in CuSCN as the HIL.  The manuscript is presented in a neat form with detailed information about sample preparation and data collection. The results are clearly presented and discussed so the overall opinion about the manuscript is a good one. I believe it can be considered for publication with some minor observations:

  1. The results in Table 1 should be discussed more clearly. There is a trend change for 0.03% doping ratio so a possible explanation for it can be provided.
  2. The conclusions section points out more the reason of choosing 0.02% F4TCNQ doping ration than the advantages of F4TCNQ-doped CuSCN is a potential HIL. I believe this fragment should be moved in the Results and discussion section and the conclusions should focus on its advantages over the currently used technologies and methods.

Author Response

  1. We thank the reviewer’s comments. From Table 1 It is seen that the performance of CuSCN-based QLEDs were enhanced with increasing F4TCNQ concentration, and the best performance was achieved with 0.02 wt% F4TCNQ. The device performance was then decreased with a higher F4TCNQ doping ratio of 0.03 wt%. We speculate that imbalanced electron-hole transport might occur when overdosed F4TCNQ was incorporated. Besides, high doping concentration has been reported to cause structural changes and energetic disorder in the host semiconductor, which reduces the efficiency and brightness of the QLEDs. The above description has been added in the main text and two references #22, 35 have been cited here.
  2. We thank the reviewer’s suggestion and the fragments about SEM, AFM, UPS, and XPS results have been moved in adequate parts in the Results and discussion. The conclusion has been re-written to focus on the advantages of F4TCNQ doping in CuSCN, including smoother surface, higher hole mobility and more balanced carrier transport as compared with the pristine one. Compared to currently used HILs such as organic PEDOT:PSS or inorganic nickel oxide, F4TCNQ-doped CuSCN has better stability against moisture and can be processed at lower temperatures, respectively. The above description has been added in the revised manuscript.

Reviewer 2 Report

This manuscript reports a potential of CuSCN as a hole injection layer in quantum dot light-emitting diodes. The authors introduce an electron-withdrawing molecule (F4TCNQ) in CuSCN, improving hole transport properties of CuSCN. As a result, the authors demonstrate QLEDs with a brightness of 169,000 cd m-2 and a current efficiency of 35.1 cd A-1. Overall, this manuscript is well organized, but some points listed below should be clarified and fixed for publication.

  1. To compare binding energies in XPS data, XPS peaks should be calibrated using a reference point (typically the C1s peak). In this work, the authors need to use the other XPS standards (e.g., metal elements) because the samples contain carbon element. The authors did not mention about calibration in this manuscript, so the authors should provide how XPS peaks are calibrated.
  2. In Figure S2, there seems to be two slopes in the high-binding energy region for UPS spectrum of F4TCNQ-doped CuSCN, the authors should check this.
  3. The authors claim that F4TCNQ-doped CuSCN showed the improved hole transport property compared with pristine CuSCN. The authors should provide J-V characteristics of QLEDs in Figure 4.
  4. Title in Supplementary is different with that in the manuscript.

Author Response

  1. After consulting with the XPS operator, we realized that the XPS peaks were calibrated using the C1s peak (adventitious carbons) at 284.4 eV. The above description has been added in the 2.3 Characterization methods. Similar protocol has also been applied in the reference #43.
  2. We thank the reviewer’s comment. In fact, the sample F4TCNQ-doped CuSCN was characterized by UPS technique twice and both results looked similar. It is obvious that the EF calculation from a smaller slope (binding energy cutoff ~17 eV) is not convincing. As for the reason to this phenomenon, we speculate that outer electrons are not easy to escape due to strong electron-withdrawing ability of F4TCNQ that cause dragging in UPS spectra.
  3. We thank the reviewer’s suggestion and J-V characteristics of QLEDs have been provided as Figure 4(b) in the revised manuscript. It should be noted that the improved hole transport ability of F4TCNQ-doped CuSCN was verified from I-V measurements of hole-only devices. The smaller current densities of QLEDs based on F4TCNQ-doped CuSCN relative to the control device indicate that electron transport is suppressed. Similar observation has also been reported in the previous literature #22. The above description has been added in the last paragraph in the 3.2 Device fabrication.
  4. We apologize for this mistake. The title in Supplementary Information has been corrected. 

Round 2

Reviewer 2 Report

I believe that the manuscript has been sufficiently improved to warrant publication in Nanomaterials.